# Characterization of Human, Ovine and Porcine Mesenchymal Stem Cells from Bone Marrow: Critical In Vitro Comparison with Regard to Humans

**DOI:** 10.3390/life13030718

**Published:** 2023-03-06

**Authors:** Elisa Katja Westerkowsky, Adriana Marisa Soares de Almeida, Michael Selle, Oliver Harms, Katrin Bundkirchen, Claudia Neunaber, Sandra Noack

**Affiliations:** 1Hannover Medical School, Department of Trauma Surgery, Carl-Neuberg-Straße 1, 30625 Hannover, Germany; 2Clinic for Small Animal Medicine, University of Veterinary Medicine Hannover, Bünteweg 9, 30559 Hannover, Germany

**Keywords:** MSC, animal model, human, sheep, pig, differentiation, migration, adhesion

## Abstract

For research and clinical use of stem cells, a suitable animal model is necessary. Hence, the aim of this study was to compare human-bone-marrow-derived mesenchymal stem cells (hBMSCs) with those from sheep (oBMSCs) and pigs (pBMSCs). The cells from these three species were examined for their self-renewal potential; proliferation potential; adhesion and migration capacity; adipogenic, osteogenic and chondrogenic differentiation potential; and cell morphology. There was no significant difference between hBMSCs and pBMSCs in terms of self-renewal potential or growth potential. The oBMSCs exhibited a significantly higher doubling time than hBMSCs from passage 7. The migration assay showed significant differences between hBMSCs and pBMSCs and oBMSCs—up to 30 min, hBMSCs were faster than both types and after 60 min faster than pBMSCs. In the adhesion assay, hBMSCs were significantly better than oBMSCs and pBMSCs. When differentiating in the direction of osteogenesis, oBMSCs and pBMSCs have shown a clearer osteogenic potential. In all three species, adipogenesis could only be evaluated qualitatively. The chondrogenic differentiation was successful in hBMSCs and pBMSCs in contrast to oBMSCs. It is also important to note that the cell size of pBMSCs was significantly smaller compared to hBMSCs. Finally, it can be concluded that further comparative studies are needed to draw a clear comparison between hBMSCs and pBMSCs/oBMSCs.

## 1. Introduction

As early as the late 1960s, plastic-adherent cells in the bone marrow were discovered, which are known to us today under the disputable name of “mesenchymal stem cells” [1]. Mesenchymal stem cells (MSC) are multipotent, self-renewable cells with the ability to differentiate into various cell types such as chondrocytes, adipocytes and osteoblasts [2]. Consequently, these cells play an important role in many studies of repair and reconstruction of various tissues due to their diverse possibilities [3]. MSCs are not only able to differentiate into a wide variety of specialized cells, but they also have a diverse range of origins. Bone marrow, adipose tissue, umbilical cord, muscle, synovium and placenta are some of the origins from which the cells are isolated [4]. Among those, bone marrow is the most productive and easily accessible source of mesenchymal stem cells. Therefore, BMSCs are most popularly used in clinical research [3].

In order to be able to transfer novel therapies to patients, these must be tested in large animal models in addition to in vitro experiments in the laboratory. Further development of stem cell therapy is also driven by the limitations of current treatment options for various medical problems in different animal species [5]. The choice of the large animal model depends on the purpose and the comparability to humans [6].

The sheep as an animal model has gained considerable appeal in recent years and is particularly popular in the field of cartilage replacement material testing. According to Adamzyk et al., orthopedic implants are also commonly tested in sheep because they are anatomically similar to humans in structure and size of the musculoskeletal system [7]. Therefore, many studies on cruciate ligament replacements in sheep have been carried out [8]. Furthermore, the fracture healing together with the mineral composition of the bone in sheep is comparable to that of humans [9,10]. Regulation of bone metabolism [11] and investigation of bone diseases such as osteoporosis [12] are also well studied in sheep models [13]. Haddouti et al. [6] and Kalaszczynska et al. [14] conducted the first studies which compared hMSCs and oMSCs from bone marrow and adipose tissue.

The pig has also become an attractive animal model for humans due to its genetic characteristics. Although the bone is less similar to humans due to its plexiformity [15], the gastrointestinal tract, the cardiovascular system [16] and the metabolism are more similar to humans than those of ruminants [17]. Furthermore, similarities between porcine and human MSCs have been described in the literature [18]. Porcine MSCs (pMSCs) can be successfully isolated from the same sources as in humans and are comparable, both ex vivo and in vivo [18]. Moreover, the genetic characteristics are given and an immunomodulatory capacity of the pMSCs is also detectable [19]. Therefore, pigs have been used as a model for testing stem cells for the treatment of spinal cord injury. In this context, examination of the neurological and histological features of this model revealed parallels to spinal trauma in human patients [20].

For isolation, expansion and characterization of human MSCs, Dominici et al. established the minimal criteria for defining multipotent stem cells, which were proposed by the International Society for Cellular Therapy (ISCT). These include three uniform criteria. Cells are only considered as MSCs if they grow plastic adherent in cell culture and express CD105, CD73 and CD90, and they must be able to differentiate in vitro into osteoblasts, adipocytes and chondroblasts. In addition, they must not express surface molecules such as CD45, CD34, CD14 or CD11b, CD79α or CD19, and HLA-DR [21].

For sheep [5,6,20,21] and pigs [19,22,23,24,25], still many different protocols exist for the isolation and characterization of species-specific MSCs. Whether the ISCT criteria can also be applied to these species is unclear so far. In 2020, Haddouti et al. compared human mesenchymal stem cells to those of sheep from three corresponding sources. MSCs derived from adipose tissue, femoral marrow fat and bone marrow were studied. The MSCs revealed solid growth behavior and strong immunomodulatory capacity for both species and each source in comparison. In addition, common positive (CD29, CD44, CD73, CD90, CD105, CD166) and negative (CD14, CD34, CD45) surface markers were identified for human and sheep. Moreover, gene expression of RUNX2 and Col2a1 during osteogenic differentiation was comparable to that in humans and revealed a slower mineralization process in ovine MSCs compared to hBMSCs [6]. To our knowledge, the ISCT criteria were tested with porcine stem cells by Noort et al. Here, pMSCs derived from bone marrow were used. Comparison showed that pMSCs express surface antigens also found on hMSCs (CD90, MSCA-1 (TNAP/W8B2 antigen), CD44, CD29 and SLA class I). In addition, hMSCs and pMSCs are comparable in adipogenic, osteogenic and chondrogenic differentiation, although pMSCs formed fat much faster than hMSCs [18].

Therefore, the aim of this study was to determine whether porcine or ovine BMSCs are more similar to those of humans in order to be able to decide on a large animal model which can be used most meaningfully, e.g., in a novel non-union treatment using BMSCs. The lack of standardized techniques for isolating and purifying stem cells remains the major limitation in research across animal species. So far, stem cells have only been used experimentally to treat a variety of diseases in different animal species. Researchers would like to switch from the experimental route to the standardized route, which is why such studies are necessary in large numbers [5].

There is a need to understand the full spectrum of stem cell effects and the preclinical evidence for safety and therapeutic efficacy, as significant gaps in clinical knowledge are apparent even in the more advanced research on hBMSCs [20]. For this reason, this project comprehensively compared the stem cell properties of sheep and pigs in vitro with those of humans under the same standardized culture conditions to assess cell size, t self-renewal, adhesion and migration potential, as well as differentiation into osteoblasts, adipocytes and chondrocytes.

## 2. Materials and Methods

### 2.1. Donor Data Recruitment

The human-bone-marrow-derived mesenchymal stem cells (hBMSCs) were obtained from voluntary donors at the Department of Trauma Surgery at Hannover Medical School. After receiving extensive information, the donors consented to bone marrow donation during an elective surgery under anesthesia by iliac crest puncture. The study protocol and process of sample donation complied with the Declaration of Helsinki, and the ethics committee of Hannover Medical School (Votum No. 2562) gave ethical approval. For the present experiment, three donors were randomly selected who met ISCT criteria in previous experiments carried out in the Department of Experimental Trauma Surgery at Hannover Medical School. The porcine-bone-marrow-derived mesenchymal stem cells (pBMSCs) and ovine-bone-marrow-derived mesenchymal stem cells (oBMSCs) were obtained from three animals each euthanized during other animal experiments at the Institute for Laboratory Animal Science (Animal testing application 20/3417) of the Hannover Medical School and as part of a cadaver study at the small animal clinic of the University of Veterinary Medicine Hannover. All donor samples used were collected within 15 min of death and were plastic-adherent. Bone marrow was harvested from the iliac crest in all donors of all species. The human donors were female and aged between 18 and 20 years. The pig and sheep donors were female and approximately three to four years old.

### 2.2. BMSCs Isolation and Cultivation

The extraction and isolation of pBMSC and oBMSC was performed as described before for human BMSC [26]. In brief, a sterile skin incision was made at the iliac crest, followed by insertion of the Jamshidi™ bone marrow puncture needle. The bone marrow was aspirated under maximum suction into the syringes coated with heparin. Samples were stored at 4 °C in a NaCl–heparin mixture for a maximum of 24 h until BMSC isolation.

Afterwards, bone marrow was diluted 1:3 with Phosphate-Buffered Saline (PBS; Dulbecco (#L1825), Biochrom, Berlin, Germany) and separated utilizing a synthetic polysaccharide–epichlorohydrin copolymer (Biocoll^®^, Biochrom) by centrifugation for 30 min at 500× *g* without brake. This resulted in the typical gradient phases of plasma, mononuclear cells, Biocoll and erythrocytes. The mononuclear cells, visible as a cloudy white ring, which also contained the BMSCs, were removed, washed again with PBS (Biochrom) and centrifuged with 500× *g* for 5 min with brake.

The resulting cell pellet was resuspended in Dulbecco’s Modified Eagle’s Medium (DMEM) FG0415 (Biochrom, Berlin, Germany) supplemented with 10% Hyclone^®^ Fetal Bovine Serum (FBS; Fischer Scientific, Schwerte, Germany), 20 mM 4-(2-hydroxyethyl)-1-piperazineethanesulfonic acid (HEPES), 1% (100 U/mL/100 µg/mL) penicillin/streptomycin (P/S; Biochrom) and 2 ng/mL recombinant human fibroblast growth factor 2 (FGF-2; PeproTech, Hamburg, Germany), afterwards called culture medium, and transferred to a culture bottle. Cells were incubated in an atmosphere of 5% CO_2_ at 37 °C.

### 2.3. Colony-Forming Unit–Fibroblast (CFU-F) Assays and Growth Rate

In order to assess the self-renewal potential of the cells, colony-forming unit–fibroblast assays were performed in passage 4 in this study. For this purpose, BMSCs were seeded as duplicates onto a 6-well plate (9.6 cm²) after expansion at three different concentrations (125, 250 and 500 cells per well) and incubated for 10 days at 37 °C and 5% CO_2_ in 3 mL culture medium per well. The cells were fixed with methanol and stained with 1% crystal violet for 30 min. After washing with distilled water, colonies were counted and the number of colonies per 100 cells was calculated [27].

To investigate the proliferation rate, the relative doubling time of the cells in the various cell passages was determined. For this purpose, a defined number of cells was seeded, and when the cells reached 70–90% confluence, they were detached using a 0.05%/0.02% trypsin–EDTA solution (Biochrom). The number of cells at the end of this passage was determined using a Neubauer counting chamber. The following formula was used to calculate doubling time as previously described by Selle et al. [28]:$$\mathrm{Doubling}\;\mathrm{Time}\;(T_{d})=\frac{\ln\left(2\right)\cdot d}{\ln\left(\frac{N_{d}}{N_{0}}\right)}$$
where N_0_ = cells seeded, N_d_ = cells counted at the end of the passage and d = days in culture.

### 2.4. Adhesion Assay

The adhesion assay was used to test how many cells adhere to a specific surface (polystyrene, #655180 96 well culture plate, Cellstar) after a defined period of time. For this, a cell suspension with 10,000 cells in 100 µL of culture medium was required per well. In addition, an internal standard with 1000, 2500, 5000, 7500 and 10,000 cells in sixfold determination were applied. Furthermore, a well without cells as a blank control was filled with 100 µL of culture medium. Six wells per donor per time point were each filled with 100 µL of cell suspension (time point t = 0 min). The plate was then incubated at room temperature. After 10, 20, 30 and 60 min, the medium was carefully removed from one experimental series of the 96-well plate and carefully washed with PBS (Biochrom). Then, 100 µL of fresh culture medium was added to the wells. After 60 min (completion of the last time point), 10 µL of WST-1 reagent was quickly pipetted into each well. The plates were then placed in the incubator and incubated at 37 °C and 5% CO_2_. After 90 min, the measurement of absorbance at 450 nm on the Microplate Spectrophotometer (BioTek Instruments, Sursee, Switzerland) was performed and corrected with the absorbance at 630 nm. Afterwards, the blank value was subtracted and the values for each group were averaged.

### 2.5. Migration Assay

This assay was performed to estimate the migration behavior of adherent cells in medium with 1% of serum, here called “deficiency medium”. For this purpose, a standardized gap of 500 µm was created in a cell lawn by use of “ibidi inserts” (96 Well Culture Plate, Thermo Fisher Scientific, Schwerte, Germany); it was assumed that the cells would migrate into it to close it. In each compartment of the ibidi chambers, 10,000 cells were seeded in 70 µL of culture medium. The experiment was performed in triplicate and incubation was performed for 24 h at 37 °C and 5% CO_2_. The ibidi inserts were carefully removed afterwards and 3 mL of deficiency medium was added per well. For each scratch, the same four locations were photographed after 12 and 24 h. The cell-free area of the scratch was evaluated using the program ImageJ and calculated as the percentage related to the overgrown area with the Wound Healing Size Tool.

### 2.6. Differentiation

Differentiation in the adipo-, osteo- and chondrogenic directions was performed in passage 4. For adipo- and osteogenesis, 150,000 cells per 9.6 cm² (6-well) were seeded and incubated at 37 °C and 5% CO_2_ in culture medium. For chondrogenic differentiation, cell pellets were formed in conical tubes of 250,000 cells by centrifugation for 5 min at 200× *g* and were then incubated at 37 °C and 5% CO_2_ in culture medium. The next day, the medium was changed into the respective specific differentiation medium.

For osteogenic differentiation, DMEM FG0415 containing 0.1 µM dexamethasone (Sigma-Aldrich, Taufkirchen, Germany), 50 µM ascorbate-2-phosphate (Sigma-Aldrich), 3 mM di-sodium hydrogen phosphate (Merck), 20 mM HEPES, 10% FBS and 1% Penicillin/Streptomycin (P/S) was used. For adipogenic differentiation, DMEM FG0435 (Biochrom) containing 1 µM dexamethasone, 60 µM indomethacin (Sigma-Aldrich), 500 µM 3-isobutyl-1-methylxanthine (Sigma-Aldrich), 10 µg/mL insulin (Sigma-Aldrich), 20 mM HEPES, 20% FBS and 1% P/S was used. The control medium for osteogenesis and adipogenesis consisted of DMEM FG0415 supplemented with 20 mM HEPES, 10% FBS and 1% P/S.

For chondrogenesis, the differentiation medium consisted of DMEM FG0435 supplemented with 20 mM HEPES, 1% P/S, 0.1 μM dexamethasone, 10 μL/mL insulin/transferrin/selenium (Sigma-Aldrich), 170 µM ascorbate-2-phosphate, 1 mM sodium pyruvate (Biochrom), 350 μM proline (Carl Roth, Karlsruhe, Germany) and 10 ng/mL transforming growth factor beta-3 (TGF-β3; PeproTech). The control medium did not contain TGF-β3.

On days 0 and 27, the differentiations in osteo- and adipogenesis were stopped. For this, the medium was removed, the wells were washed with 2 mL of PBS (Biochrom) and fixed with 1 mL of 4% formalin for 30 min. Afterwards, the washing was repeated, this time with distilled water.

Chondrogenesis was stopped on day 27 and the chondrogenesis pellets were fixed for histology. Washing and fixation procedures were identical for the 6-well plates. After fixation, the pellet was frozen with Tissue Tek (Tissue Tek OTC Blue, Sakura Finetek, Torrance, CA, USA) embedded, and 5 µm thick cryosections were made (CM 3050S, Leica Biosystems, Wetzlar, Germany).

Calcium ions from osteogenic differentiation that reside in the extracellular mineralized matrix of osteoblasts were stained for 10 min with alizarin red (Roth, 0.5% dissolved in distilled water, pH 4.5). Adipocytes were stained with Oil Red O (Sigma-Aldrich, 5 g/L, dissolved in 60% (*w*/*v*) isopropanol) for 25 min. The chondrogenesis pellets were stained with Safranin O for 15 min to analyze glycosaminoglycans in the cartilage. The degree of osteogenic and chondrogenic differentiation was determined by calculating the percentage of stained area relative to total area. For a valid result, three representative images of the respective populations were taken and evaluated.

All analyses were performed blindly without prior knowledge of underlying donor data and with the same settings. For the evaluation of the osteogenesis and adipogenesis, slides of representative images were taken with the light microscope (Olympus CKX41), which were used for all evaluations in this paper, at a magnification of 10 and evaluated using a self-written tool from the Department of Experimental Trauma Surgery of the Hannover Medical School. For the chondrogenesis, the Keyence digital Microscope VHX-7000 was used.

### 2.7. Fluorescence

Cell suspensions of 10,000 and 30,000 cells per mL of culture medium were prepared. For each batch, 3 × 100 µL were seeded into one well of a 96-well plate and incubated for 24 h at 37 °C and 5% CO_2_. The medium from the wells was removed and washed with PBS (Biochrom). Then, 4% formalin was added to the wells and fixed for 20 min at room temperature. Washing was repeated and 0.1% Triton-X-100 (Sigma-Aldrich) in PBS was added for 3 min. The supernatant was removed and washed with PBS. This was followed by adding 100 µL of Phalloidin-iFluor 488 Conjugate (1000X, 2 µL of Stock Solution (1000X) (Biomol, Hamburg, Germany) to 2 mL 1% BSA in PBS (BSA = Albumin bovine Fraction V, pH 7.0, SERVA Electrophoresis, Heidelberg, Germany), which was incubated for 60 min at RT in the dark. After washing with PBS, 100 µL of 2 µM DAPI (Biomol) (4′,6-Diamidino-2-Phenylindole, Dihydrochloride, cell) was added for 5 min and incubated at room temperature in the dark. Images were evaluated using a mercury vapor lamp on the microscope (Olympus CKX41). DAPI staining becomes optimally visible with an exposure time of 80–140 ms, and Phalloidin-iFluor 488 Conjugate ranges from 1.25 to 2.5 s. The cell size was evaluated using the program ImageJ. To evaluate the results of fluorescence, the length and area of 10 cells were measured and compared using the Measure tool of the program ImageJ-win64.

### 2.8. Statistics

The statistical data analysis in this work was performed with the program GraphPad Prism version 9.3.1. The data analysis of the samples showed a normal distribution. Accordingly, group comparisons were performed with one-way ANOVA with Tukey’s post hoc test for correction of multiple comparisons. Grouped data were analyzed via two-way ANOVA with Sidak’s post hoc test for correction of multiple comparisons. The graphical representation of the results is shown as scatter dot plots.

Descriptive *p*-values were determined. A *p*-value less or equal to 0.05 was considered as a statistically significant difference. In the statistical evaluation of the results, the mean (M) and standard deviation (SD) are given. Biological and technical replicates are denoted with “n” and “N”, respectively.

## 3. Results

BMSCs from humans, sheep and pigs were compared in terms of their morphology (fluorescence microscopy), stem cell characteristics (CFU-F assay), growth behavior (proliferation capacity) and differentiation ability in the direction of osteogenesis, chondrogenesis and adipogenesis.

### 3.1. Fluorescence Microscopy

As shown in Figure 1, hBMSCs had a shape with thin, long processes. The length of the hBMSCs ranged between 90 and 150 µm (M: 122 µm; SD 1255.23 µm). The oBMSCs were slightly broader in shape and their length ranged from 90 to 170 µm (M = 133 µm; SD: 1022.66 µm). The pBMSCs were small and more round compared to the other species. The length of the pBMSCs ranged between 30 and 70 µm (M = 51 µm; SD: 606.17 µm). The nuclei were of comparable sizes in all three species with a diameter of about 14 µm. The area comparison is shown in Figure 2. The mean area of hBMSCs was 2105.85 µm^2^, that of pBMSCs was 592.82 µm^2^ and that of oBMSCs was 2444.94 µm^2^. pBMSC are significantly smaller (*p* = 0.004, Figure 2).

### 3.2. Colony-Forming Unit–Fibroblast Assays and Growth Rate

Evaluation of the CFU assay in Figure 3 shows that pBMSCs with a mean of 7.07% had a comparable renewal potential to hBMSCs with 6.27% (*p* = 0.84). With 1.73%, oBMSC could only form a third of the colonies compared to hBMSC (*p* = 0.01).

The doubling time in days was comparable in all species in passages 3 to 6 (*p* ≥ 0.05), whereby oBMSCs showed a wide variance in passages 4–7 (Figure 4). In passage 7, oBMSCs required a significantly higher doubling time than hBMSCs (*p* = 0.01) (Figure 4).

### 3.3. Adhesion Assay

The adhesion capacity of all three species at four different time points is shown in Figure 5. After 10 min, hBMSCs with 20.9% (SD 3.58%) were significantly different from both pBMSCs with 1.21% (SD 2.1%) (*p* = 0.006) and oBMSCs with 2.1% (SD 3.14%) (*p* = 0.006). In general, hBMSCs are found to adhere faster than oBMSCs and pBMSCs.

After 20 min, hBMSCs reached a mean adhesion of 67.06% (SD 6.1%), whereas oBMSC adhesion improved to 14.55% (SD 11.5%) and pBMSC adhesion was still low with <2% (SD 0.95%). There was once again a significant difference between pBMSCs (*p* = 0.004) and the oBMSCs (*p* = 0.01) compared to hBMSCs.

After 30 min, the adhesion of hBMSCs increased steadily and reached a mean of 84.28% (SD 19%). The adhesion of oBMSCs also increased to 24.47% (SD 15.51%) (*p* = 0.03) compared to hBMSC. The mean value of adherence of pBMSCs increased to 3.4% (SD 3.06%) (*p* = 0.03) compared to hBMSC.

After one hour, the adherence of the hBMSCs was comparable to the 30 min time point. The adherence of oBMSCs reached 31.86% (SD 21.1%) after 1 h, whereas the adhesion of pBMSCs remained low with 4.17% (SD 3.85%) (*p* = 0.04) compared to hBMSCs. After 24 h, the BMSCs from all three species were adherent.

### 3.4. Migration Assay

After 12 h, the migration capacity of oBMSCs with 12.17% (SD 8.9%, *p* < 0.001) and pBMSCs with 25.14% (SD 5.8%, *p* = 0.002) was significantly lower compared to hBMSCs with 44.67% (SD 14.4%).

After 24 h, the migration capacity of both oBMSCs with 44.8% (SD 16%) and pBMSCs with 50.9% (SD 6.6%) increased compared to the time point 12 h (Figure 6). However, it was still significantly lower compared to hBMSC with 90.2% (SD 11.5%). For oBMSCs and pBMSCs, a significant reduction in the migration capacity with *p* ≤ 0.001 was measurable compared to hBMSCs.

### 3.5. Differentiation

#### 3.5.1. Osteogenic Differentiation

Representative pictures of alizarin red staining showing mineral depositions of calcium are shown in Figure 7A on day 27 of differentiation of hBMSCs, oBMSCs and pBMSCs. All donors of each species were able to induce osteogenic differentiation (Figure 7). On day 27, 75.6% (SD 21.7%) of the area of hBMSCs, 92.5% (SD 3.7%) of the area of oBMSCs and 97.9% (SD 4.3%) of the area of pBMSCs were positively stained with alizarin red. These results indicated a better osteogenic differentiation capacity of oBMSCs (*p* = 0.03) and pBMSCs (*p* = 0.006) compared to hBMSCs. As expected, the non-induced samples showed no signs of osteogenic cell differentiation.

#### 3.5.2. Adipogenic Differentiation

Figure 8 shows the differentiation properties of the stem cells of the compared species towards adipogenesis. Each species was visibly suitable for adipogenic differentiation, but due to technical difficulties in washing off the cells, only a qualitative statement was possible. As expected, the non-induced samples showed no signs of adipogenic cell differentiation.

#### 3.5.3. Chondrogenic Differentiation

Representative pictures of chondrogenic differentiation after 27 days are shown in Figure 9A. Quantitative evaluation revealed a chondrogenesis level with 16.54% in hBMSCs (SD = 3.95%) and 23.16% in pBMSCs (SD = 13.4%) on day 27. Only 0.18% were stained with Safranin O in the oBMSCs (SD = 0.2%). These results indicate no significant chondrogenic differentiation between hBMSCs compared to pBMSCs and oBMSCs. Only hBMSCs and pBMSCs were able to perform a chondrogenic differentiation in pellet culture in all three donors. In sheep, no differences were observed between the pellets incubated in control medium or chondrogenic medium. As expected, the non-induced samples showed no signs of chondrogenic cell differentiation.

## 4. Discussion

Since cell-based methods are being used more frequently in innovative medicine, the aim of this study was to determine whether porcine- or ovine-bone marrow-derived mesenchymal stem cells, which were isolated and cultured with the established hBMSC protocol, are more similar to human-bone-marrow-derived mesenchymal stem cells in order to select an appropriate animal model for preclinical research.

The results showed that ovine BMSCs have a similar cellular appearance to hBMSCs with adequate osteogenic and adipogenic differentiation potential, but an impaired chondrogenic differentiation potential. The time to adhere to the cell culture plate and the migration time were prolonged for oBMSCs compared to hBMSCs. Porcine BMSCs have a different cellular appearance with a smaller and more round cell body, but suitable osteogenic and chondrogenic differentiation capacity. The adhesion capacity is slower than oBMSCs and slower compared to hBMSCs, whereas the migration capacity is faster than oBMSCs but slower compared to hBMSCs. Apart from this study, there is no experimental setup with all three species under the same culture conditions for direct comparison [18,20,29]. Consistent with our study, Schweizer et al. showed that cell sizes in fluorescence staining between pBMSCs and hBMSCs were significantly different and hBMSCs had a cell body almost twice as large [26]. Furthermore, Noort et al. showed that there was no difference in growth potential between hMSCs and pMSCs over a 3-week period [18]. Cells of both species were cultured in a different basal medium (medium 199) with dextrose as the carbohydrate source, to which an endothelial cell growth factor and heparin were added.

In our study, the carbohydrate source was glucose, and in addition, FGF2 was used as a growth factor and no heparin was added [18]. Despite differences in the medium, the results of our study were confirmed regarding the pBMSCs. Between hBMSCs and oBMSCs, a significant difference was only visible from passage 7. However, this finding should be interpreted with caution due to the high variability of the individual data from this passage. In order to be able to make an exact statement on this point, repetitions in late passages with several donors are necessary. Rentsch et al. showed that the doubling time of sheep MSCs is 1.2-fold higher than hBMSCs [30]. A comparable medium was used as in the present work and the doubling time was about 50 h calculated on average over five passages. In contrast, we could only show in passage 4 that the mean of the doubling time of the sheep was higher than that of the human.

Cell migration is an important function that plays a major role in tissue repair processes [31]. In this work, migration assay was used to compare the stem cells of different species in terms of migration with the same environmental factors. In our study, a less significant difference was found between the animal species compared to hBMSCs. This could be related to the results of the adhesion assay, as hBMSCs also performed significantly better than oBMSCs and pBMSCs in the migration assay. However, since the adhesion and migration assays were performed under conditions designed for the cultivation of human BMSC, the results may be different with customized media tailored to the needs of oBMSCs or pBMSCs. Osteogenic differentiation was significantly increased in oBMSCs and pBMSCs compared to hBMSCs in our study. These results are in line with the publication of Haddouti et al., who demonstrated a strong osteogenic potential of oBMSCs as early as 21 days, which showed a slower mineralization process than hBMSCs, under conditions comparable to our osteogenesis protocol and using the same staining with alizarin red [6]. The literature also includes studies that have tested the osteogenic potential on so-called 3D scaffolds. However, these in vivo experiments have shown that oBMSCs not only differentiate osteogenically on more materials than hBMSCs, but also form a higher percentage of bone than hBMSCs [29]. Noort et al. examined osteogenic differentiation of pBMSCs under three different protocols that are for humans, in which differentiation was successful in most donors [18].

After completion of adipogenic differentiation and staining, hBMSCs had more stained fat vacuoles compared with oBMSCs and pBMSCs in the current study. Especially for pBMSCs, this might be due to the fact that fat vacuoles are secreted during differentiation, and consequently, vacuoles are no longer fixed intracellularly. Thus, wash-off of the stained vacuoles could have happened during the staining and washing process. This observation is in agreement with a study performed with pBMSCs [18]. The reason for this could additionally be excessive lipid formation leading to many unfixed vacuoles. In this context, Noort et al. showed that pMSCs formed fat much faster than hMSCs [18]. For oBMSCs, there are only publications that provide a successful adipogenic differentiation but do not report about the morphology of the fat vacuoles. Confirmation of the success of adipogenesis in sheep was additionally provided by the study of McCarty et al. Again, positive results were obtained after four weeks in induced medium intended for human stem cells. However, the number of stained fat vacuoles was also low, which is consistent with our study [32]. Regarding the chondrogenic differentiation, our study showed that pBMSCs are comparable to hBMSCs. This is in line with the findings of Noort et al., supporting the findings that pMSCs are able to differentiate in the chondrogenic direction. However, chondrogenic differentiation in oBMSCs was not successful in our study. In turn, Haddouti et al. showed that oMSCs in pellet culture can perform chondrogenesis after 21 days. They cultured them on agarose gel and stained with Alcian Blue, and the medium was comparable to the one used in our study [6]. In addition, Zscharnack et al. showed that chondrogenic differentiation of oMSCs is enhanced under low oxygen conditions (5% O_2_) [33]. Furthermore, mechanical stimulation showed the improvement of chondrogenesis potential [34].

## 5. Limitations

As this study deals with donated hBMSCs and BMSCs from pigs and sheep which were sacrificed due to other reasons, some limitations must be considered. We used BMSCs from healthy, middle-aged patients in this study, which cannot reflect the complex human situation in a clinical setting with different comorbidities. Apart from that, the age and sex of all human and other BMSC donors were not taken into consideration. In addition, the conditions chosen for BMSC isolation and cultivation were optimally designed for hBMSCs. There is a lack of uniform protocols for working with oBMSCs and pBMSCs that are optimal for the cells and can be applied uniformly across laboratories. Further, the site of collection and the characteristics of the donor of MSCs are of great importance to the research results and complicate the standardization of protocols.

## 6. Conclusions

The results on the properties of stem cells from sheep and pigs in terms of in vitro self-renewal, growth, adhesion and migration potential, as well as differentiability and cell size, showed various differences compared to hBMSCs.

Based on cell morphology, oBMSCs are more comparable to hBMSCs, while pBMSCs are more comparable to hBMSCs based on successful chondrogenic differentiation and their self-renewal potential. Under the standardized culture conditions defined here, various differences between species are thus present, so caution should be exercised in in vitro studies when directly transferring oBMSCs and pBMSCs to humans. However, no clear recommendation can be made; thus, further comparative studies are necessary.

## Figures and Tables

**Figure 1 life-13-00718-f001:**
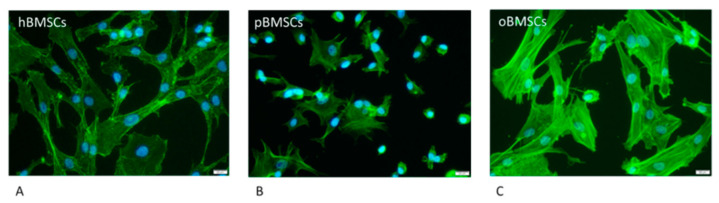
Morphology of the different BMSCs. The BMSCs of the different species were stained with Phalloidin-iFluor 488 conjugate to stain the actin filaments within the cytoplasm (green) and DAPI to stain the nucleus (blue) The scale is 20 µm and the cells were in passage 3. (**A**) hBMSCs; (**B**) pBMSCs; (**C**) oBMSCs.

**Figure 2 life-13-00718-f002:**
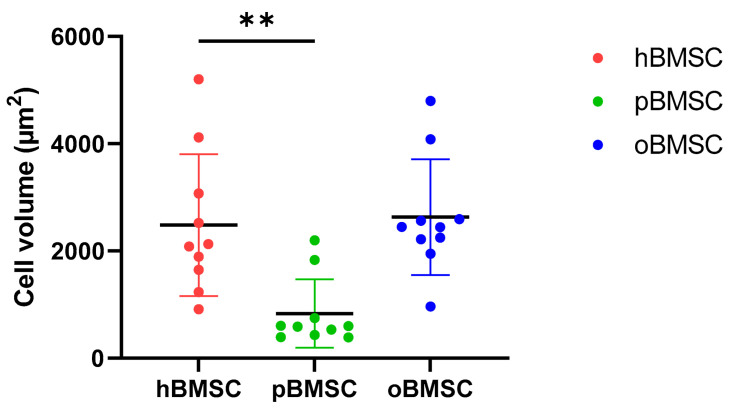
Cell volumes of BMSCs. The different cell volumes are shown in µm^2^. The volumes of hBMSCs and oBMSCs are approximately comparable. The pBMSCs are only one-third of the size of the hBMSCs. N = 10, ** *p* < 0.002.

**Figure 3 life-13-00718-f003:**
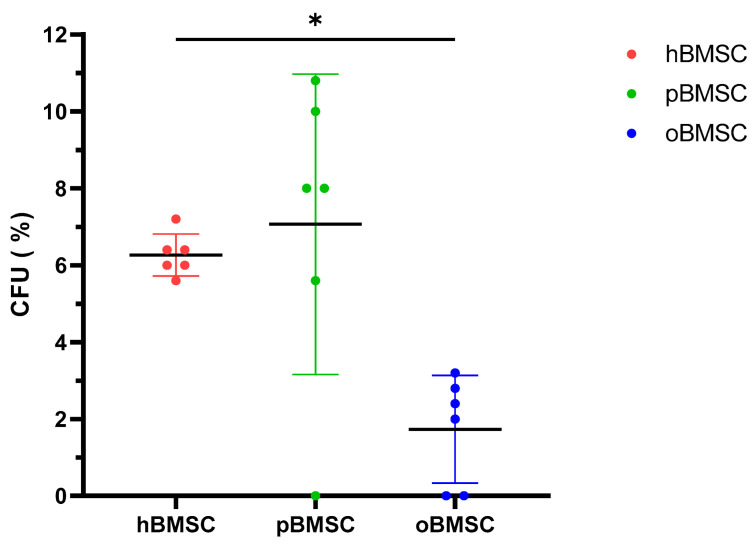
Colony-forming unit–fibroblast assay (CFU-F). The CFU-F assay was performed in P4 of hBMSCs, pBMSCs and oBMSCs. Percentage of CFU-F of the three experimental species (n = 3, N = 2). A significant lower percentage of CFUs was detected in oBMSC compared to hBMSC (* *p* = 0.01).

**Figure 4 life-13-00718-f004:**
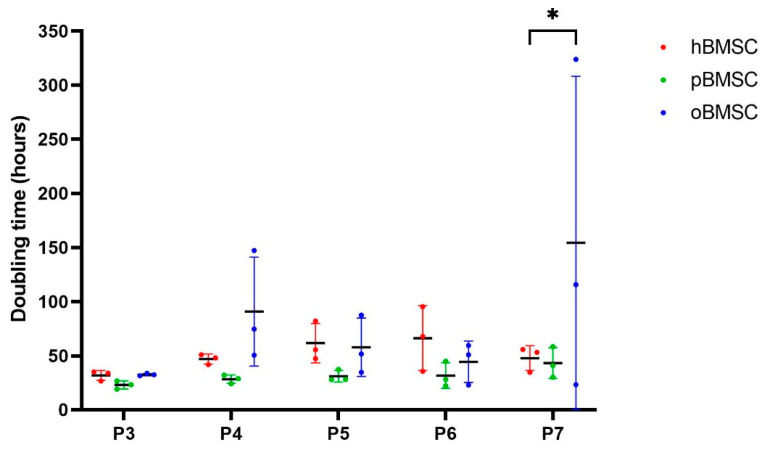
Growth rate. The growth rate shown is represented by doubling time in hours of human, sheep and porcine BMSCs over passages 3 to 7. A significant difference was found between oBMSCs and hBMSCs in passage 7 (* *p* = 0.01, n = 3).

**Figure 5 life-13-00718-f005:**
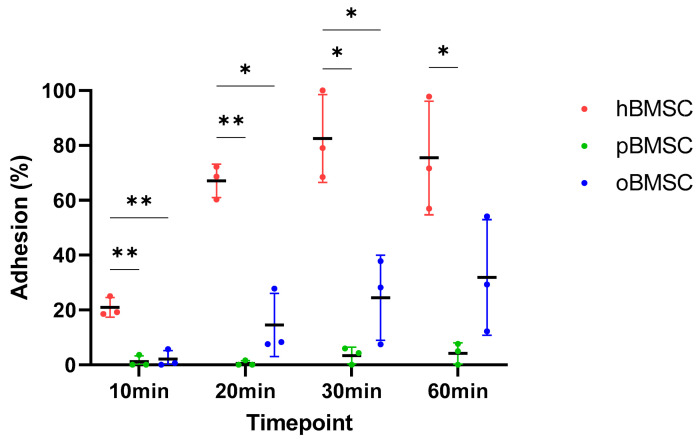
Adhesion assay. The percentage of the adhesion was analyzed and compared between hBMSCs, oBMSCs and pBMSCs after 10, 20, 30 and 60 min. At all time points, the adhesion of hBMSCs were significantly higher compared to oBMSCs and pBMSCs. n = 3, * *p* < 0.033, ** *p* < 0.002.

**Figure 6 life-13-00718-f006:**
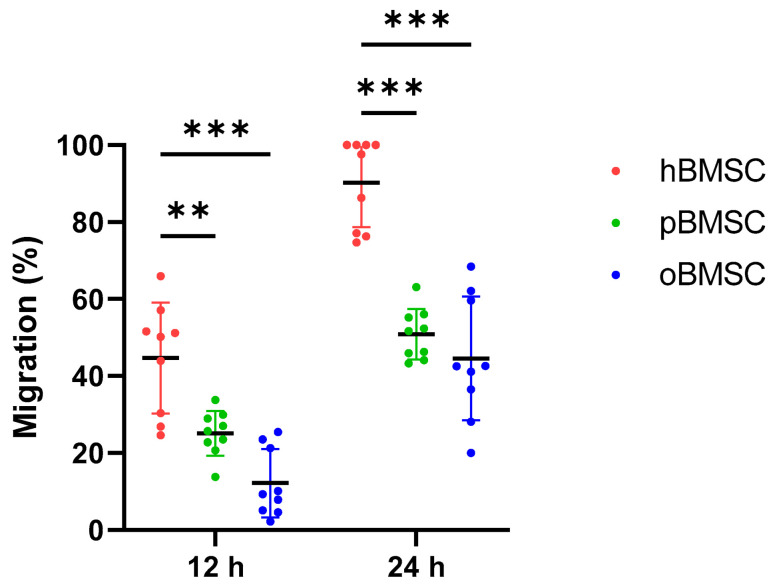
Migration assay. Degree of migration (%) in oBMSCs and pBMSCs compared to hBMSCs after 12 and 24 h. Significant differences between the two animal species and hBMSCs were measurable after both time points, where hBMSCs migrated the fastest. n = 3, N = 3 ** *p* < 0.002, *** *p* < 0.001.

**Figure 7 life-13-00718-f007:**
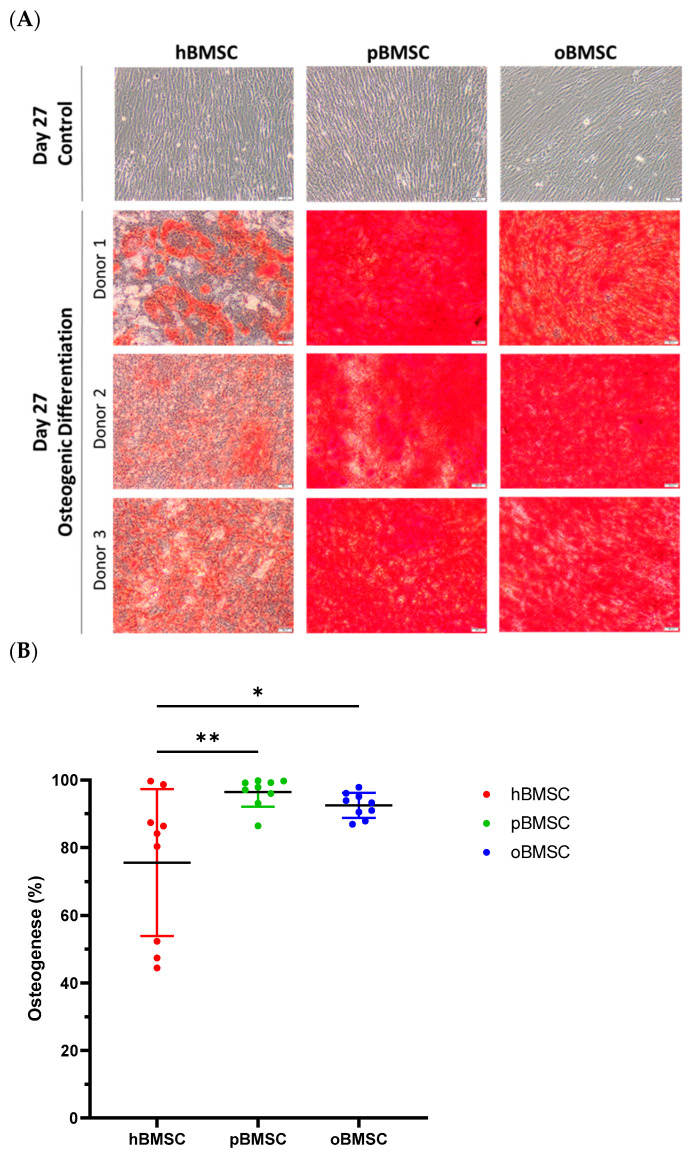
Osteogenic differentiation. (**A**) Representative images of osteogenic differentiation of hBMSC, pBMSCs and oBMSC of three different donors. Undifferentiated cells as control on day 27 are shown in the first row. Pictures after osteogenic differentiation stained with alizarin red on day 27 are shown in the other rows. Scale bar = 50 µm. (**B**) Osteogenic differentiation was evaluated after 27 days by alizarin red staining. hBMSCs showed a significantly smaller stained area than oBMSCs and pBMSCs. n = 3, N = 3 * *p* < 0.033, ** *p* < 0.002.

**Figure 8 life-13-00718-f008:**
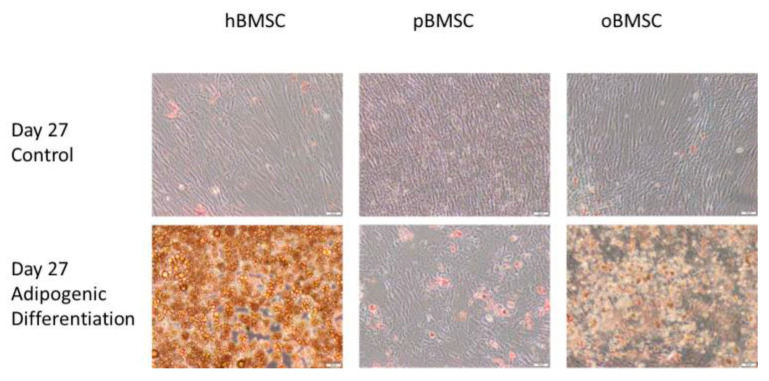
Adipogenic differentiation. Representative images of adipogenic differentiation after staining with Oil Red O of hBMSC, pBMSCs and oBMSCs. Controls are shown above and pictures of induced adipogenic differentiation are shown below. Little to no adipogenesis was observed in the control. Scale bar = 50 µm.

**Figure 9 life-13-00718-f009:**
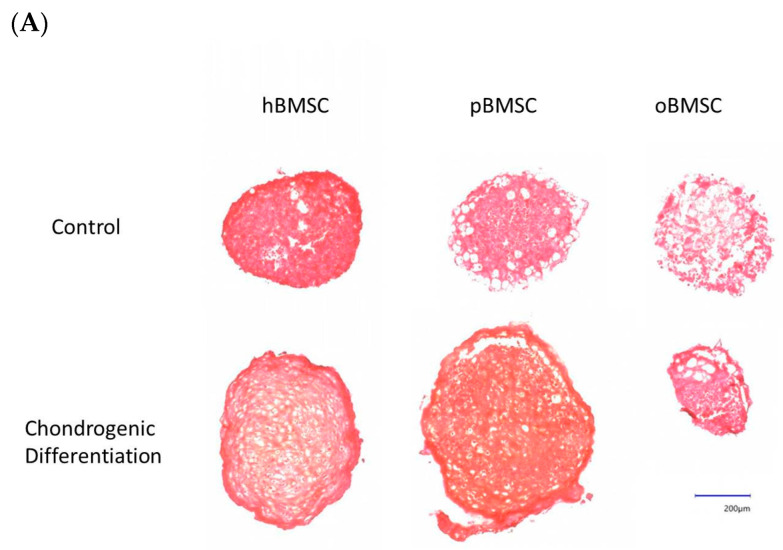
Chondrogenic differentiation. (**A**) Representative images of chondrogenic differentiation of hBMSCs, pBMSCs and oBMSCs. Chondrogenic differentiation was performed for 27 days. The first line shows the pellets incubated with control medium. The lower pictures show the pellets 27 days after chondrogenic induction. The scale is 200 µm. Only hBMSCs and pBMSCs show signs of chondrogenic differentiation in the induced medium. Scale bar = 200 µm (**B**) Chondrogenic differentiation was evaluated after 27 days by Safranin O. There were no significant differences between pBMSCs and oBMSCs to hBMSCs, n = 3.

## Data Availability

The data used to support the findings of this study are available from the corresponding author upon request.

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
