# Peer review of "Characterization of Human, Ovine and Porcine Mesenchymal Stem Cells from Bone Marrow: Critical In Vitro Comparison with Regard to Humans"

_life, 2023, doi:10.3390/life13030718_

Round 1
Reviewer 1 Report
This is comparative study in vitro that aims side by side to compare human, pig and sheep BMSCs morphology, growth behaviour and multipotent differentiation under same culture conditions designed for human BMSCs. Comparative studies using cells from humans and other animal species are not novel, but may be of interest for cell biologists and researchers searching for an optimal animal model in various tissue regeneration studies.
Results presented in the manuscript represent only in vitro behaviour of human, pig and sheep MSCs isolated from bone marrow. As we well know, in vitro results does not necessarily correlate with in vivo results. Although results have certain value, they could not be translated into clinical setting and remain purely basic.
Manuscript is well and clearly written, albeit methods and presentation of some results could be improved. For example, calculation of cell proliferation and doubling time is not clearly described. It would be much clearer if authors can provide graphic cell proliferation results over time (10 days) and in the table indicate cell doubling time expressed in hours instead of days.
The results of chondrogenic differentiation presented in the manuscript are not convincing. Both, control and experimental pellets look positive for Safranin O, although authors point that control samples showed no signs of chondrogenic differentiation. In this disputable case, I suggest additional staining for Alcian blue or maybe evaluation for chondrogenic gene expression.
There are some mistakes in the discussion part. On line 398-399, authors state: The adhesion and migration capacity (of pBMSCs) are slower than oBMSCs and slower compared to hBMSCs. In fact, figure 4 shows that migration of pBMSCs is faster compared to oBMSCs.
Author Response
Reviewer 1:
Manuscript is well and clearly written, albeit methods and presentation of some results could be improved. For example, calculation of cell proliferation and doubling time is not clearly described. It would be much clearer if authors can provide graphic cell proliferation results over time (10 days) and in the table indicate cell doubling time expressed in hours instead of days.
Answer: We have expanded the description of the doubling time in the methods part. We have also converted the values ​​given in days into hours and replaced the table with a graphic to optimize the display.
The results of chondrogenic differentiation presented in the manuscript are not convincing. Both, control and experimental pellets look positive for Safranin O, although authors point that control samples showed no signs of chondrogenic differentiation. In this disputable case, I suggest additional staining for Alcian blue or maybe evaluation for chondrogenic gene expression.
Answer: Thanks for the hint. In fact, due to the overall rather low chondrogenic differentiation of the samples used in this manuscript, no major difference in safranin O staining between control and induced samples is observed. In the case of an optimally progressed chondrogenic differentiation - as shown in the attached example here - a red-purple coloration can be seen in the control pellets due to Safranin O. The structure of the pellet shows a diffuse network and the overall pellet is smaller than that of the induced, chondrogenically differentiated pellets. As soon as chondrogenic differentiation of the stem cells has taken place, the staining of safranin O changes to an orange-red hue. Furthermore, the typical chondrocyte cavities can be observed in the structure of the pellet.
To illustrate this, we created this figure using other hBMSC samples from our laboratory:
From our experience, Alcian Blue is not helpful in this case. Here, too, a coloration can be seen in the control pellets and the difference in structure can primarily be seen.
Originally, we also wanted to carry out a gene expression analysis. However, we decided against it due to the lack of established assays for sheep and pigs.
There are some mistakes in the discussion part. On line 398-399, authors state: The adhesion and migration capacity (of pBMSCs) are slower than oBMSCs and slower compared to hBMSCs. In fact, figure 4 shows that migration of pBMSCs is faster compared to oBMSCs.
Answer: Thanks for the hint. We have changed the sentence accordingly.

Reviewer 2 Report
The submitted article compares in a clear and organized way the cell morphology, self-renewal potential, proliferation potential, adhesion and migration capacity, adipogenic, osteogenic and chondrogenic differentiation potential, of three different sources of bone marrow-derived mesenchymal stem cells: human, ovine and porcine. This study aims to obtain useful information for the use of stem cells in research and clinical trials.
Despite the overall good presentation of the results, there are some remarks that need to be into account;
Majors:
-the authors claim that in passage 7, oBMSCs required a significantly higher doubling time than hBMSCs. However, the astonishing variability on the data at that specific time point (0.97, 13.49, 4.82) makes consider the results carefully. I suggest either repeat the assay or/and include a later passage to confirm the assumption.
- Osteogenic differentiation is evaluated after 27 days, when the process is so advance that differences are not outstanding. Would it not be better to evaluate also another shorter time point?
-The technical difficulty is the adipogenic differentiation assay was due to the fact that fat vacuoles are secreted during differentiation and consequently vacuoles are no longer fixed intracellularly. Could not be wiser to do shorter differentiation time to evaluate a timepoint where vacuoles are already there and not secreted yet so the process can be analyzed?
-Images of chondrogenic differentiation assay of hBMSCs, pBMSCs and oBMSCs do not clearly show what is expected. Better images, higher magnifications, clearer controls and a better description of what is expected to see will be helpful.
Minors:
- Acronyms should be detailed the first time that they appeared (oBMSCs, hBMSCs and pBMSCs)
- This message appears several times and I don´t understand the meaning: “Error! Reference source not found.”
Author Response
Reviewer 2:
-the authors claim that in passage 7, oBMSCs required a significantly higher doubling time than hBMSCs. However, the astonishing variability on the data at that specific time point (0.97, 13.49, 4.82) makes consider the results carefully. I suggest either repeat the assay or/and include a later passage to confirm the assumption.
Answer: That's a valid objection. In our view, however, examining another later passage is not helpful. The reason for this is that by passage 8 many cells stopped proliferating and even died. This could be recognized by the increasing doubling time. The cells needed more and more time to double. Therefore, because of the senscence, an analysis in higher passages makes little sense. The focus must be on earlier passages.
We have included a corresponding sentence in the discussion for explanation:
“However, this finding should be treated with caution due to the high variability of the individual data from this passage. In order to be able to make an exact statement on this point, repetitions in late passages with several donors are necessary.”
- Osteogenic differentiation is evaluated after 27 days, when the process is so advance that differences are not outstanding. Would it not be better to evaluate also another shorter time point?
Answer: It is actually the case that, in retrospect, shorter differentiation times would have made sense. Unfortunately, this was not possible in this approach due to the small number of available bone marrow and correspondingly few pBMSCs and oBMSCs. We therefore decided to differentiate the endpoint on day 27.
-The technical difficulty is the adipogenic differentiation assay was due to the fact that fat vacuoles are secreted during differentiation and consequently vacuoles are no longer fixed intracellularly. Could not be wiser to do shorter differentiation time to evaluate a timepoint where vacuoles are already there and not secreted yet so the process can be analyzed?
Answer: Yes, that's right. Shorter times would definitely have made sense. Alternatively, regular photo documentation of the cells during differentiation, since the fat vacuoles are clearly visible in adipogenic differentiation even without staining. In the end, the reason for this was the insufficient number of stem cells coupled with the fact that differentiation from fat cells plays a subordinate role in trauma surgery and the focus is more on bones and cartilage.
-Images of chondrogenic differentiation assay of hBMSCs, pBMSCs and oBMSCs do not clearly show what is expected. Better images, higher magnifications, clearer controls and a better description of what is expected to see will be helpful.
Answer: Unfortunately, we do not have any images with a lower magnification. Nevertheless, we have created a new figure presenting images of osteogenic differentiation from three different donors of each of the three different species. However, here, too, it was shown that the osteogenic differentiation for the species pig and sheep is very strong overall. Due to this strong staining, a focused recording was difficult and this is what leads to the impression that the recordings look artificial. Nevertheless, these correspond to reality.
Formularende
Minors:
- Acronyms should be detailed the first time that they appeared (oBMSCs, hBMSCs and pBMSCs)
Answer: Thanks for the hint. We have changed it.
- This message appears several times and I don´t understand the meaning: “Error! Reference source not found.”
Answer: Thanks for the hint. We have corrected the error.

Reviewer 3 Report
This is an interesting study characterizing the differences between human, pig, and sheep bone marrow derived mesenchymal stem cells. The cellular characterization includes different cell morphologies, differentiation potential, cell migration and proliferation differences. It is interesting to see the differences of these three different species, however, I have major issues with the meanings of this study in a big picture, including the usefulness of this study. I do acknowledge authors describe some; however, this point should be emphasized more in the introduction clearly. It will be nice to have some examples of what the author thinks in using the MSC from different species, and what they think is useful for this detailed characterization.
Minor points are summarized below.
1) It is nice to have the full meaning of abbreviation at the first appearance, so readers do not need to go back and forth to look for the actual meaning in the later part of the manuscript. (i.e. hBMSC, pBMSC, and oBMSC in the abstract)
2) There are multiple “Error! Reference source not found” all over the paper.
Author Response
Reviewer 3:
It is interesting to see the differences of these three different species, however, I have major issues with the meanings of this study in a big picture, including the usefulness of this study. I do acknowledge authors describe some; however, this point should be emphasized more in the introduction clearly. It will be nice to have some examples of what the author thinks in using the MSC from different species, and what they think is useful for this detailed characterization.
Answer: An attempt was made to further emphasize the urgency of this study by adding a few more sources. The following two sources have been added:
Voga M, Adamic N, Vengust M and Majdic G (2020) Stem Cells in Veterinary Medicine—Current State and Treatment Options. Front. Vet. Sci. 7:278. doi: 10.3389/fvets.2020.00278 :
Harding, J., Roberts, R.M. & Mirochnitchenko, O. Large animal models for stem cell therapy. Stem Cell Res Ther 4, 23 (2013). DOI: https://doi.org/10.1186/scrt171
In addition, the description in the introduction has been expanded:
“Further development of stem cell therapy is also driven by the limitations of current treat-ment options for various medical problems in different animal species [5].” […]
“The lack of standardized techniques for isolating and purifying stem cells remains the major limitation in research across animal species. So far, stem cells have only been used experimentally to treat a variety of diseases in different animal species. Accordingly, one would like to switch from the experimental route to the standardized route, which is why such studies are necessary in large numbers [5].
There is a need to understand the full spectrum of stem cell effects and the preclinical evidence for safety and therapeutic efficacy, as significant gaps in clinical knowledge are apparent even in the more advanced research on hBMSCs [22].”
1) It is nice to have the full meaning of abbreviation at the first appearance, so readers do not need to go back and forth to look for the actual meaning in the later part of the manuscript. (i.e. hBMSC, pBMSC, and oBMSC in the abstract)
Answer: Thanks for the hint. We have changed it.
2) There are multiple “Error! Reference source not found” all over the paper.
Answer: Thanks for the hint. We have corrected the error.

Round 2
Reviewer 2 Report
Thanks for the explanation of my comments
Author Response
We also thank you for the constructive comments.